

# Assessing the impact of Syrian refugees on earthquake casualty estimations in southeast Turkey

Bradley Wilson[1] and Thomas Paradise[1]

[1]Department of Geosciences, University of Arkansas, Fayetteville, Arkansas

*Correspondence to:* Bradley Wilson (bsw006@uark.edu)

**Abstract.** The influx of millions of Syrian refugees into Turkey has rapidly changed the population distribution along the Dead Sea Rift and East Anatolian fault zones. In contrast to other countries in the Middle East where refugees are accommodated in camp environments, the majority of displaced individuals in Turkey are integrated into local cities, towns, and villages— placing stress on urban settings and increasing potential exposure to strong earthquake shaking. Yet, displaced populations are

often unaccounted for in the census based population models used in earthquake casualty estimations. Accordingly, this study constructs a refugee inclusive gridded population model and analyzes its impact on semi-empirical casualty estimations across southeast Turkey. Daytime and nighttime fatality estimates were calculated for five geographically distributed fault zones at earthquake magnitudes 5.8, 6.4, and 7.0. Total casualty estimates ranged from 28-7723 individuals, with the contribution of refugees varying from 1%-26% of total estimated casualties. On average, these percentages correspond to casualty underesti-

mations of tens to hundreds of individuals. These findings communicate the necessity of incorporating refugee statistics into earthquake risk analyses in southeast Turkey and the ongoing importance of placing environmental hazards in their appropriate regional and temporal context.

## 1 Introduction

Since Syria's devolution into Civil War in early 2011, millions of Syrians have fled into Turkey seeking reprieve from areas of

territorial conflict. As of December, 2016, the refugee population in Turkey is nearing 2.8 million, with majority populations located in southeastern provinces (Republic of Turkey, 2015). This influx of population has rapidly changed the population distribution of earthquake prone areas near the East Anatolian and Dead Sea Rift fault systems, increasing the number of individuals potentially exposed to strong earthquake shaking.

     The refugee crisis in Turkey is unique in several ways that amplify earthquake risk concerns. In contrast to other countries

in the Middle East, the majority of Syrian refugees in Turkey are settled amongst local populations rather than formalized refugee camps. Therefore, migrated populations represent significant increases in population density—a form of temporary urbanization. Rapidly increasing urban populations is stressing local cities seeking to adequately house refugee populations (3RP, 2015).

     In the past, high rates of urbanization have been attributed to the Turkish government's failure to enforce seismic building

codes (Erdik, 2001). The structural integrity of existing building stock is a widespread issue throughout Turkey (Ilki and Celep,





2012), with poor code enforcement contributing to high death tolls in recent earthquakes (Erdik, 2001; Güney, 2012). Fatality occurrence in earthquakes is strongly linked to building collapse (Oskai and Minowa, 2001; Nadim et al., 2004; Coburn and Spence, 2002). This linkage contributes to the disparaging earthquake mortality rates in the developing versus the developed world. Earthquake resistant structures are both expensive to construct and time consuming to license and verify, creating

opportunities for corrupt payments, bribes, and a lack of political incentives to drive the under enforcement of building codes (Keefer et al., 2011; Anbarci et al., 2005). The combination of poor historical precedent for earthquake mitigation and rapidly increasing occupancy in urban structures raises concerns about the scale of future earthquake disasters in southeast Turkey.

Structural vulnerability is critically intertwined with assessments of population exposure in earthquake risk analyses. Accurately mapping population exposure is an essential part of the risk analysis process for environmental hazards (Chen et al.,

2004; Freire and Aubrecht, 2012; Aubrecht et al., 2012). The presence of Syrian refugees in southeast Turkey has complicated this process, especially as it pertains to datasets commonly used in earthquake loss estimations. Displaced Syrian populations are tracked at varying levels by the Turkish government and international agencies. Refugees are registered at the province level, but are allowed to freely move within their registered province under the Temporary Protection Regulations stipulations (Çorabatır, 2016). Thus, the position of refugees within any designation smaller than provinces—district, city, village—is

uncertain. These uncertainties present challenges for earthquake loss estimations that rely on accurate population estimates.

Improved human exposure data impacts several components of the risk analysis process, including loss estimation and disaster relief (Chen et al., 2004; Aubrecht et al., 2012; Guha-Sapir and Vos, 2011). Studies by Aubrecht et al. (2012) and Ara (2014) have shown the paramount importance incorporating temporal factors into population datasets. Despite these findings, most earthquake related hazard studies do not account for temporal population changes and instead rely on census-based

population estimates (Freire and Aubrecht, 2012). In the absence of building level data on structural class and time-varying occupancy (which are often nonexistent, especially in developing nations), casualty estimations rely on using census data or modified versions of census data—either disaggregated by uniformly distributing population over an aeral unit or converted into a finer-resolution dasymetric model using a variety of geographical constraints.

Casualty estimation tools play an important role in both mitigation and relief and recovery processes. Because loss estimation

systems can be used to direct rapid post-event humanitarian decisions (Jaiswal et al., 2011b), their accuracy is crucial. In Turkey, refugee populations are not accounted for in the census data due to recency—the last census was completed in the 2011, the same year of the Syrian crisis' onset. Therefore, any product produced using census based population sources is likely to underestimate population exposure unless explicitly adjusted for Syrian populations. In southeastern provinces where the percentage of Syrian refugees often reaches over ten percent of the native population, accounting for these populations is a

salient component of the casualty estimation process.

Accordingly, this study addresses this challenge by (1) minimally modeling refugee populations statistics with Turkish population estimates into a gridded population dataset and (2) assessing the corresponding impact on earthquake casualty estimations across four geographically distributed fault segments across southeast Turkey. Using the semi-empirical loss estimation technique of Jaiswal and Wald (2010), human casualty estimates are simulated for a range of earthquake magnitudes. By com-

paring the casualty estimates from the refugee inclusive population model to traditional census based approaches, it is shown



what magnitude of underestimations should be expected in the absence of dedicated population exposure adjustments. These results communicate why incorporating migration statistics is a necessary step for properly assessing earthquake exposure in this region—communicating the ongoing importance of analyzing natural hazards within an appropriate regional setting.

## 2   Study area

As of December, 2016, there are 2,790,767 registered Syrian refugees in Turkey, over half of the Syrian conflict's total refugees and more than any other country in the Middle East. Turkey currently has twenty three refugee camps operating at full capacity across ten provinces, accommodating approximately 10% of the total registered population. The remaining  90% of refugees are settled amongst local communities in their provinces of registration. This comes in stark contrast to other countries in the Middle East where a majority of refugees are housed in camped environments. The Turkish Ministry of the Interior Directorate General of Migration Management consistently updates these statistics as more Syrians are formally registered as refugees within the country.

A majority ( 60%) of Syrian refugees have settled in southeastern provinces near the Turkey-Syria border, with the highest concentrations located in provinces bordering Syria directly (Fig. 2). The area of focus for this study encompasses twelve primary southeastern provinces and portions of three additional provinces. This region extends from the northwest corner of Kayseri to the southeast corner of Şanlıurfa (Fig. 1). Tectonically, this region is dominated by two primary left lateral strike-slip fault systems, the East Anatolian fault zone and the Dead Sea Rift fault zone, which bound the intersection between the relatively stable Arabian platform and the Anatolide-Tauride block. The precise structural relationship between these two fault systems is complex and poorly understood. Their intersection is generally placed at a triple junction near the city of Kahramanmaraş (Chorowicz et al., 1994), or slightly further south near Antakya (Over et al., 2004). Various explanations for the mechanics of the two systems have been explored in Doğan Perinçek and İbrahim Çemen (1990); Duman and Emre (2013). Under either explanation, refugee settlement in southeastern Turkey represents a migration away from a stable tectonic setting into an area characterized by frequent earthquake activity.

## 3   Historical seismicity

There is a wealth of information detailing the long history of earthquake activity on the East Anatolian and Dead Sea Rift fault systems (Ambraseys, 2009; Sbeinati et al., 2005; Barka and Kadinsky-Cade, 1988; Garfunkel et al., 1981). Ambraseys (2009) provides a detailed overview of historical seismicity in the region, with Sbeinati et al. (2005) providing additional information on Syrian earthquakes. Both the East Anatolian and Dead Sea Rift fault systems have seen a recent quiescence in seismic activity, but paleoseismic evidence indicates a consistent long term pattern of infrequent large earthquakes (Ambraseys, 1989; Meghraoui et al., 2003). Large earthquakes in southern Turkey first appear in the historical records in 148 B.C.E. in the writings of John Malalas, who chronicles the destruction of the city of Antioch due to the 'Wrath of God', a phrase often used to describe earthquake events (Ambraseys, 2009). The city of Antioch alone, located in the modern day Hatay province, is



Natural Hazards
and Earth System
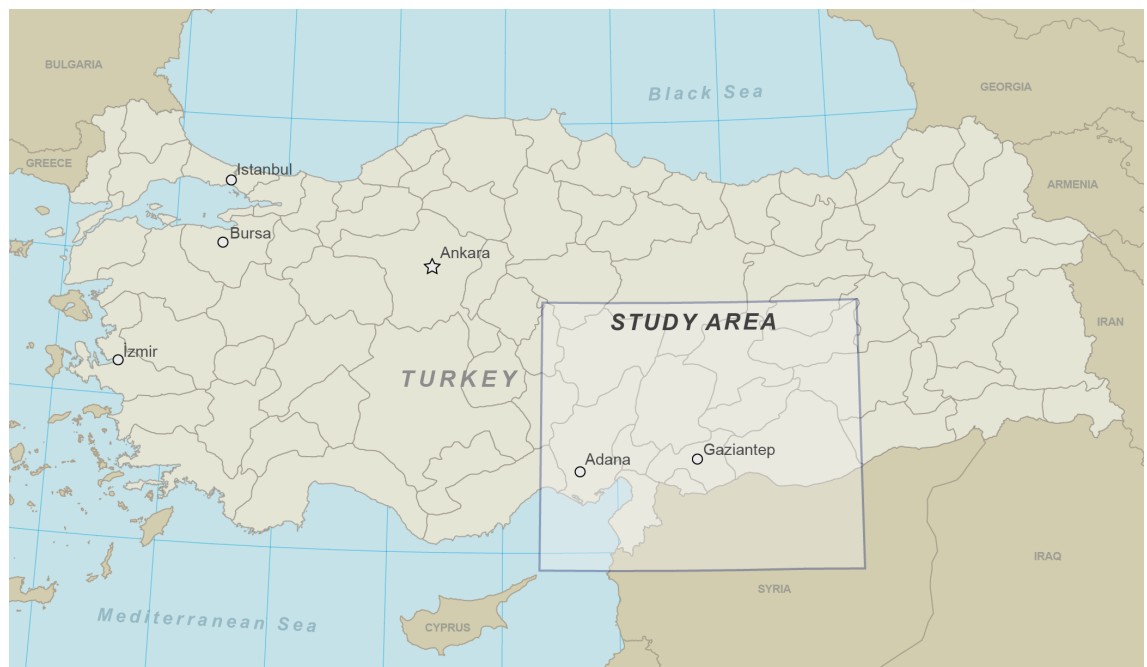

**Figure 1.** Study area within southeastern Turkey.

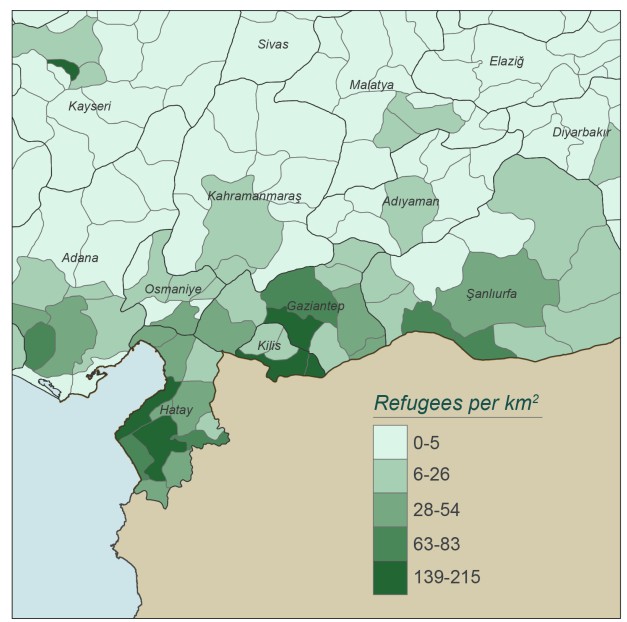

**Figure 2.** Migrated population density, December 2016.



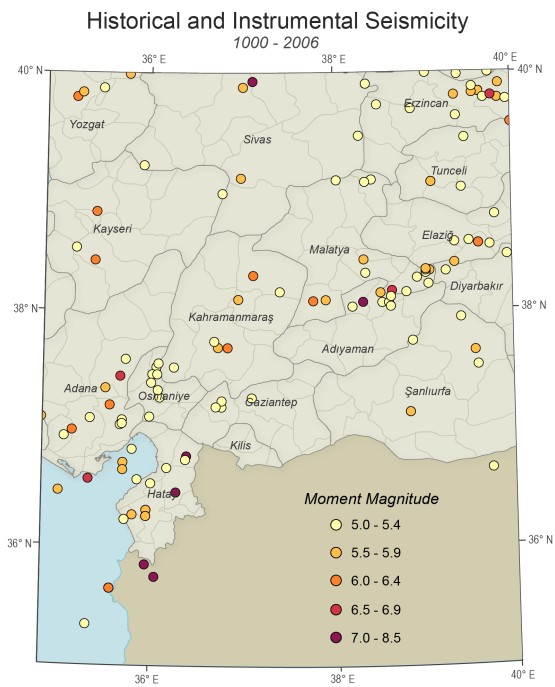

**Figure 3.** The distribution of earthquake shaking as gathered from historical documents and modern seismic networks, compiled in (Sesetyan et al., 2013).

shaken over forty times before 1900 C.E.. Figure 3 plots seismic activity greater than magnitude 5.0 across the study area over the last millennia, showing an fairly even distribution across the length of the fault zones.

Historical records also provide insight into the human impact of several notable earthquakes. The earthquakes that destroy Antioch in 115 C.E. and 526 C.E. are estimated to have killed 250,000 or more individuals each. If these numbers are correct, both earthquakes fall into the top ten most deadly earthquakes of all time (Musson, 2001) (the death estimates may be exaggerated, but are generally considered to be plausible (Ambraseys, 2009)). The 526 C.E. earthquake is particularly notable, striking on the 29th of May, Ascension Day. Ambraseys (2009) notes that the high death totals for this earthquake (250,000-300,000) were likely amplified by the influx of visitors into the city.

# 4 Data and methods

## 4.1 Refugee-inclusive population model

The last Turkish census was completed in 2011 before the onset of Syrian mass migration. Therefore, most population models built from census-based sources do not account for the presences of Syrian refugees. This is not an intentional error (The





Gridded Population of the World, version 4 dataset (GPWv4) (Doxsey-Whitfield et al., 2015) explicitly states this particular shortcoming), but rather a systematic problem associated with infrequent data collection. Any forward modeled population dataset for Turkey based on pre-2011 data will mischaracterize true populations in high migration areas unless refugees are explicitly included. Furthermore, the uncertainty associated with the sub-province level position of refugees complicates their

inclusion in disaggregated grid-based datasets. Population models that include Syrian migration do exist, most notably Oak Ridge National Laboratory's LandScan$^{TM}$ database (ORNL, 2016)—a proprietary product used by the U.S. Department of Defense and U.S. Geological Survey among others.

As a framework for modifying regional census data for inter-period migration events, a geographic information systems (GIS) workflow was utilized to construct a regional refugee inclusive gridded population model using freely available data

from Turkey's Address Based Population Registration System (ABPRS) and the Turkish Directorate General of Migration Management. This model, like the GPWv4, employs a minimally modeled aeral distribution process that disaggregates administrative population counts into cells of equal population. Turkish district level boundaries from the GADM database of Global Administrative Areas (GAA, 2015), clipped to the area of interest, were first converted into three kilometer grid cells and equally distributed 2015 ABPRS populations according to the proportional number of cells in each district.

Refugee migration data is monitored at the province level, one administrative boundary larger than the ABPRS estimates. As mentioned above, the exact position of non-camped Syrian refugees within their respective provinces is unknown. Accordingly, the existing district level population distribution was used as a proxy for refugee settlement patterns. The non-camped refugee population was distributed according to each district's relative percentage of its corresponding province province percentage. Known camped refugee populations were assigned to their residing district and removed from the populations otherwise dis-

tributed. The model was finalized by repeating the process used above for distributing ABPRS populations to allocate refugees into equally populated grid cells. The resulting gridded population model (Fig. 4) is spatially consistent and has discrete values for base population and migrated population.

### 4.1.1 Advantages and drawbacks

This population model has a number of advantages and drawbacks over other freely available population models. The primary

advantage being that, in contrast to other aeral grid models, this model explicitly accounts for the spatial distribution refugee migration, as characterized by the Turkish Directorate General of Migration Management. Furthermore, it takes into account the most recently available statistical information from the Turkish government regarding local population counts. The 2015 ABPRS data does provide slightly improved estimates compared to 2011 census data forward modeled with growth equations. Large study areas or regions with smaller migrated populations might prefer a globally gridded model (like the GPWv4

database), but such datasets suffer in this particular region due to the scale of short period population changes. Proprietary gridded population models like LandScan$^{TM}$ are updated yearly and may provide improved characterization of refugee settlement, but their dasymetric mapping approaches are not open source.

Two primary assumptions were necessary for the construction of this model. First, refugees were distributed from the province level to the district level in an equivalent distribution to the existing population. It is probable that actual refugee pop-





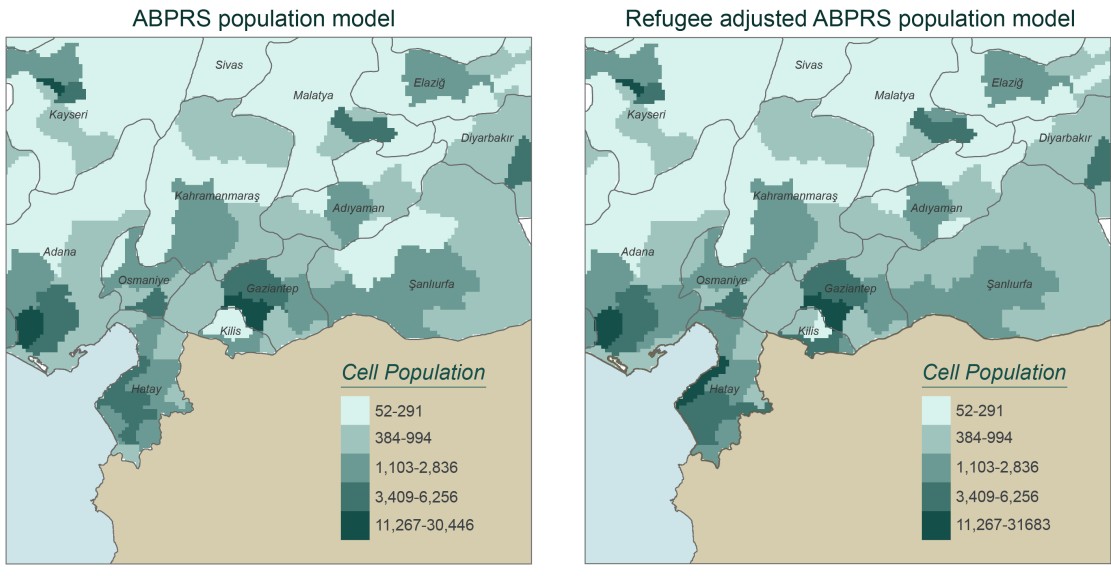

*Both models gridded at 3 km resolution*

**Figure 4.** Gridded cellular population models produced from Turkish ABPRS data before (left) and after (right) including refugee statistics.

ulations exhibit different spatial clustering patterns, but this information is difficult to model without additional constraining information. The Turkish government has started tracking province level registration and implemented freedom of movement restrictions as of August, 2015, but these regulations still allow for movement for refugees within their registered provinces (Çorabatır, 2016). Assuming an equivalent population distribution to that of existing populations maintains the regional urban-
5  rural breakdown—an important element in building-type assignment for loss estimation. Outside of specific refugee camp locations (which have been accounted for), there is not clear evidence for assigning an alternative distribution pattern. Secondly, refugee populations were aerally distributed equally into district level grid cells—the same assumption made for Turkish populations. This assumption retains consistency between population types in the absence of sufficient reason to minimally model refugees differently than that of existing population.

10  **4.2 Earthquake scenarios**

Earthquake scenarios are an important tool for emergency management planning. Tools like the USGS' Prompt Assessment of Global Earthquake Risk (PAGER) system and FEMA's HAZUS software have been used in the U.S. for emergency planning and both the national and state level (FEMA, 2008; Chen et al., 2016; EERI, 2015). As part of the earthquake loss estimation process, synthetic ground motion fields were produced for a series of earthquake ruptures spanning five faults across south-
15  eastern Turkey. For each fault, moment magnitude 5.8, 6.4 and 7.0 earthquakes were simulated. This spread of earthquake magnitudes reflects moderate to major earthquakes within the magnitude range of historical earthquakes in the area as seen in earthquake catalogs covering Turkey (Zare et al., 2014; Woessner et al., 2015). Five earthquake epicentral locations were



**Table 1.** Earthquake rupture parameters

| Fault name | Hypocenter (Lon,Lat) | Depth | Dip | Rake |
|---|---|---|---|---|
| Pütürge | (38.20, 37.77) | 13.2 | 70.0 | 0.0 |
| Kırıkhan | (36.08, 36.27) | 13.2 | 80.0 | 0.0 |
| Türkoğlu | (37.48, 37.04) | 13.2 | 80.0 | 0.0 |
| Göksun | (37.03, 35.77) | 13.2 | 80.0 | 0.0 |
| Bozova | (37.32, 38.59) | 13.2 | 80.0 | 180.0 |

Upper and lower boundaries of the seismogenic layer were set to 0 and 20 km,
respectively.

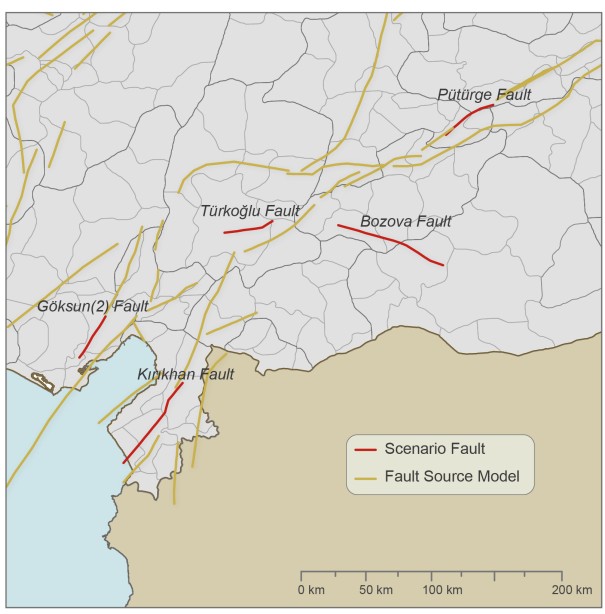

**Figure 5.** Fault locations selected for earthquake scenarios.

selected along fault traces provided in the fault-source background model in the Seismic Hazard Harmonization of Europe (SHARE) project. It should be noted that the choice of exact epicentral location is somewhat arbitrary, but can have an impact on casualty estimations. Epicenters for this study were selected to represent geographically distributed fault segments and were chosen independently of refugee migration patterns. An overview of rupture information for each fault segment is available in

5    Table 1.

The Global Earthquake Model's OpenQuake software platform was utilized to produce ground motion fields for each earthquake scenario. OpenQuake's scenario-based hazard assessment implements ground motion prediction equations to estimate the geographic distribution of shaking intensity for a user-specified fault rupture (GEM, 2016). All of our scenarios utilized





the ground motion prediction equation detailed in Akkar and Bommer (2010), relevant to earthquakes in Europe and the Middle East. We accounted for seismic site amplification using Vs30 estimates from the USGS Global Vs30 Map server, which estimates Vs30 from topographic slope (Wald and Allen, 2007). OpenQuake implements site parameters by assigning each observation grid cell the site parameters of the nearest measurement in the Vs30 grid (GEM, 2016). For each earthquake scenario, we produced ten ground-motion fields—each resampling the aleatory uncertainty in the calculation.

## 4.3 Casualty estimations

A variety of techniques exist for performing earthquake casualty estimation. Jaiswal et al. (2011b) defines three primary categories: empirical, analytical, and hybrid approaches. Empirical and analytical methodologies exist at opposite ends of the spectrum, with empirical approaches estimating casualties as a function of past losses in similar geographical areas while analytical methodologies estimate casualties using structural engineering approaches. Hybrid, or semi-empirical approaches can be thought of as simplified analytical approaches, estimating casualties using empirical estimates of structural parameters. The choice of methodology depends primarily on data availability and the scale of analysis (Jaiswal et al., 2011b).

This analysis seeks to assess the impact of refugee migration on the magnitude of earthquake casualty estimates. Fully empirical approaches that rely on fatalities in past earthquake events are poorly suited to analyzing short term variability in loss estimations, a fundamental portion of this analysis. A fully analytical approach would be preferable, but the structural engineering and building occupancy data does not exist to support such calculations. Even if structural information was available at the individual buildings level, improved refugee settlement data would be necessary for disaggregating province level refugee populations into specific building occupancy. Therefore, this study instead employs the hybrid, semi-empirical loss estimation approach detailed in (Jaiswal and Wald, 2010), given by Eq. (1).

$$E[L] \approx \sum_{i=1}^{n} \sum_{j=1}^{m} P_i \times f_{ij} \times CR_j(S_i) \times FR_j \tag{1}$$

This approach estimates fatalities given a series of n grid cells and m structural types. Each grid cell's population $P_i$ is first broken out into a fractional percentage for a given structural type $f_{ij}$. Fatalities are then calculated based on of the collapse rate of structural type j ($CR_j(S_i)$) at macroseismic intensity ($S_i$), and the fatality rate $FR_j$ of structure type j under collapse (Jaiswal and Wald, 2010).

Empirical data from the World Housing Encyclopedia (WHE)-PAGER project, phase I, was used to constrain collapse rates. Jaiswal and Wald (2009b) provides estimates of the building stock distribution under the PAGER taxonomy along with estimated collapse percentages. It is noted that several of the collapse probabilities for Turkey are listed higher than those generalized for European macroseismic scale intensities across the entire WHE-PAGER phase I dataset (Jaiswal et al., 2011a). Accordingly, when available, collapse rates are calculated using generalized fragility coefficients using Eq. (2). For building types without published coefficients, values have been estimated using the methodology in Jaiswal and Wald (2010), minimizing the residual error of the power function in Eq. (2) fit to a single set of collapse rates at given intensities. Fatality rates are implemented using values from Jaiswal and Wald (2010) for building types with HAZUS-MH fatality rates, and generalized





**Table 2.** Collapse rates and fatality rates by structural type

| Structural Class | PAGER-WHE Type | Collapse % by Intensity | | | | FR (%) |
|---|---|---|---|---|---|---|
| | | VI | VII | VIII | IX | |
| Masonry | DS | 0 | 1 | 14 | 45 | 8 |
| | A | 2 | 17 | 48 | 90 | 6 |
| | UFB | 0 | 3 | 18 | 43 | 6 |
| | UCB | 0 | 0 | 3 | 10 | 8 |
| Structural Concrete | C2 | 0 | 0 | 0 | 2 | 15 |
| | C3 | 0 | 0 | 2 | 11 | 15 |
| | C6 | 0 | 1 | 5 | 15 | 15 |
| | C7 | 0 | 2 | 22 | 45 | 15 |
| | PC2 | 0 | 1 | 6 | 15 | 15 |
| Steel | S1 | 0 | 0 | 0 | 1 | 14 |
| Wood | W | 0 | 2 | 10 | 20 | 13 |

Collapse rates are rounded to the nearest percent.

Turkish values from Porter et al. (2008) in their absence.

$$CR_j(S) = A_j \times 10^{\frac{B_j}{S - C_j}} \qquad (2)$$

The casualty estimation process was implemented by loading the average peak ground acceleration (PGA) values for each scenario into GIS software and spatially joining them to the gridded population model. The grid raster for each scenario was exported to a CSV file containing each cell's district identifier code, PGA value, pre-migration population estimate, and migrated population estimate. These CSV files were combined with the fragility information provided in Table 2 and population distribution information from Table 3 to implement Eq. (1) in R. For each grid cell, the PGA value was converted to Modified Mercalli Intensity values using the relationship of Wald et al. (1999) and the gridded cell populations were fractionally divided into building types according to the percentages in Table 2. While it is probable that the structural occupancy of refugees is different than that of local populations (approximately 25% of refugees live in makeshift or rubble housing (3RP, 2015)), estimates for this distribution were not sufficiently known and both population sources were distributed equivalently. Any cell with a population density greater than 150 persons per kilometer was assigned an urban distribution while the rest were assigned a rural distribution (OECD, 1994). Each scenario's gridded population was distributed for both daytime and nighttime percentages using the corresponding information in Table 3.





**Table 3.** Building occupancy percentages by structural type and time of day, from Jaiswal and Wald (2009b).

| Structural Class | PAGER-WHE Type | Urban Daytime | Urban Nighttime | Rural Daytime | Rural Nighttime |
|---|---|---|---|---|---|
| Masonry | DS | 4 | 15 | 0 | 1 |
| | A | 2 | 15 | 0 | 2 |
| | UFB | 25 | 35 | 15 | 35 |
| | UCB | 5 | 5 | 15 | 25 |
| Structural Concrete | C2 | 5 | 0 | 5 | 0 |
| | C3 | 40 | 25 | 50 | 36 |
| | C6 | 5 | 0 | 6 | 0 |
| | C7 | 8 | 0 | 5 | 0 |
| | PC2 | 2 | 0 | 2 | 1 |
| Steel | S1 | 0 | 0 | 1 | 0 |
| Wood | W | 4 | 10 | 1 | 1 |

Daytime refers to working hour percentages, nighttime to living hour percentages.

## 5   Results and discussion

The total number of projected casualties in a particular earthquake scenario depends on the spatial overlap between population, shaking intensity, and structural type distribution. Adjustments in any of these parameters affects the number of projected casualties. This study estimated casualties for fifteen earthquake scenarios representing three earthquake magnitudes at five geographically distributed fault zones. Casualties were calculated for both daytime and nighttime hours and refugee inclusive and refugee exclusive population scenarios, producing a total of four casualty estimates for each earthquake scenario. The results indicate intra-fault, inter-fault, and temporal differences in earthquake casualty projections across the study area, as well as notable casualty increases after refugee inclusion.

Tables 4 and 5 present the casualty numbers for each of the twelve earthquake scenarios. Table 4 provides baseline casualty estimates produced using the gridded population model before Syrian refugee adjustment, while Table 5 updates the estimations using the refugee inclusive population model. It is important to highlight that the values presented in Tables 4 and 5 are not specific casualty predictions for future events, but representations of the order of magnitude that could be expected in events of varying size and location. Therefore, any conclusions drawn henceforth are scenario specific—and should only be generalized to other scenarios with appropriate caution.

### 5.1   Consistent trends

Refugees and local populations were disaggregated using a consistent methodology across population scenarios. Accordingly, several of the trends drawn from Table 4 and 5 are consistent across every population scenarios and not tied to refugee migra-



**Table 4.** Fatality estimates prior to including refugee population statistics.

| Fault | $M_w$ | Daytime Casualties | Nighttime Casualties |
|---|---|---|---|
| Pütürge | 5.8 | 27 | 62 |
| | 6.4 | 91 | 178 |
| | 7.0 | 202 | 372 |
| Türkoğlu | 5.8 | 430 | 657 |
| | 6.4 | 945 | 1380 |
| | 7.0 | 1514 | 2187 |
| Kırıkhan | 5.8 | 1268 | 1886 |
| | 6.4 | 2832 | 3991 |
| | 7.0 | 4461 | 6144 |
| Göksun | 5.8 | 773 | 1119 |
| | 6.4 | 1712 | 2402 |
| | 7.0 | 2944 | 4099 |
| Bozova | 5.8 | 646 | 980 |
| | 6.4 | 1335 | 1942 |
| | 7.0 | 2111 | 3055 |

tion. At all five fault locations, increasing earthquake magnitude from 5.8 to 6.4 resulted in a larger casualty increase (241%
average) than the subsequent 6.4 to 7.0 magnitude increase (175% average). This reflects the non-linear relationship between
earthquake magnitude, intensity, and building collapse rates. Increases in earthquake ground motions, not magnitude, are the
driving force behind increase earthquake casualties. These results are consistent with general magnitude–intensity relation-
5  ships. Areas simulated with PGA values high enough to produce building collapse are highly localized for magnitude 5.8
scenarios compared to larger magnitudes. However, local site conditions or poorly constructed buildings can amplify casual-
ties even in moderate magnitude earthquakes—the 1960 Agadir earthquake in western Morocco resulted in 15,000 casualties
despite a moment magnitude of 5.7 (Paradise, 2005).

Nighttime casualties were forecasted consistently higher than daytime casualties in all fault locations (an average of 160%).
10  This indicates that the building stock distribution occupied during working hours is less susceptible to collapse than the build-
ing stock distribution occupied during living hours. These results follow out of Table 3 which shows populations generally
transitioning from vulnerable masonry buildings to concrete structures during working hours. These findings add to the grow-
ing volume of research stressing the importance of including temporal elements into natural hazards studies (Chen et al., 2004;
Ara, 2014; Aubrecht et al., 2012; Freire and Aubrecht, 2012; Guha-Sapir and Vos, 2011).



**Table 5.** Casualty estimates after including refugee population statistics.

| Fault | Mw | Total (Day) | Total (Night) | New (Day) | New (Night) | % Difference (Day) | % Difference (Night) |
|---|---|---|---|---|---|---|---|
| | 5.8 | 28 | 63 | 1 | 1 | 3.7 | 1.7 |
| Pütürge | 6.4 | 92 | 181 | 1 | 3 | 1.1 | 1.7 |
| | 7.0 | 205 | 377 | 3 | 5 | 1.5 | 1.3 |
| | 5.8 | 466 | 711 | 36 | 54 | 8.4 | 8.2 |
| Türkoğlu | 6.4 | 1022 | 1492 | 77 | 112 | 8.1 | 8.1 |
| | 7.0 | 1637 | 2363 | 123 | 176 | 8.1 | 8.0 |
| | 5.8 | 1594 | 2371 | 326 | 485 | 25.7 | 25.7 |
| Kırıkhan | 6.4 | 3560 | 5017 | 728 | 1026 | 25.7 | 25.7 |
| | 7.0 | 5607 | 7723 | 1146 | 1579 | 25.7 | 25.7 |
| | 5.8 | 773 | 1119 | 55 | 80 | 7.1 | 7.1 |
| Göksun | 6.4 | 1712 | 2402 | 123 | 175 | 7.2 | 7.3 |
| | 7.0 | 2944 | 4099 | 216 | 301 | 7.3 | 7.3 |
| | 5.8 | 694 | 1056 | 48 | 76 | 7.4 | 7.8 |
| Bozova | 6.4 | 1437 | 2099 | 102 | 157 | 7.6 | 8.1 |
| | 7.0 | 2285 | 3321 | 174 | 266 | 8.2 | 8.7 |

Refugee percentages represent the fractional component of the total casualties resulting from refugee inclusion.

Inter-fault casualty differences reflect the location of particular faults to population centers. The two highest casualty locations, the Kırıkhan and Göksun fault segments, extend over several of the highest population districts in all of southeast Turkey. With casualties estimates ranging from hundreds at magnitude 5.8 to thousands at magnitudes 6.4 and 7.0, these scenarios indicate serious consequences in the event of similar earthquake ruptures. The proximity of major cities and other population dense areas to faults strongly contributes to increased earthquake casualty estimates in this region. This has been historically true as well, with earthquakes repeatedly destroying ancient cities near the modern day provinces of Hatay and Adana (Ambraseys, 2009).

## 5.2 Refugee related trends

Examining the new casualty and percent difference statistics in Table 5 shows that the presence of refugees has a non-trivial impact on earthquake impacts. Consistent increases in simulated casualties reflects the widespread presence of refugees amongst Turkey's southeast districts. The varying percentages between fault locations reflects the distribution of refugee populations across the study area. Syrian refugee settlement patterns do not directly correspond to those of existing populations. While several previously populated areas in southeast Turkey (Adana and Gaziantep) have sustained similarly large refugee populations,



many refugees are settled in less populated provinces like Hatay, Şanlıurfa, and Kilis. Kilis represents the extreme end of the spectrum with approximately 120,000 Syrian refugees settled amidst an existing population of only approximately 130,000 individuals.

Understanding these variations explains the differences in the earthquake casualties by location. Table 5 shows total casual-
ties, which reflects both population sources, and new casualties that reflect the impact of refugee adjustment. The percentage difference statistic measures the relative difference between these two values. The findings show that the Kırıkhan fault scenarios sustain the largest percent increases in addition to retaining the most overall casualties. However, this pattern is not consistent. The Türkoğlu and Bozova fault scenarios show the second largest percentage increases, but have less total casualties than the Göksun fault scenarios at comparable magnitudes. The Pütürge fault remains the least deadly earthquake location
and has negligible percent increases from refugee migration.

It naturally follows that provinces with larger population increases will sustain larger percentages of additional casualties in nearby earthquakes. Yet, it had not been previously shown what casualty magnitude should be associate with current refugee populations. The results from this study indicate that casualty underestimations range from 1-26%, varying with location. These percentages translate into a wide range of casualty counts. The deadliest of the simulated earthquakes, a moment magnitude
7.0 rupturing the Kırıkhan fault during nighttime hours equated to 1579 additional casualties compared to the base population scenario estimate, while the least deadly Pütürge scenario only produced one additional casualty after refugee adjustment. For magnitude 6.4 earthquakes and above, refugee inclusion adds hundreds of casualties to the total event values at four of five fault locations, a notable severity increase.

It is noted that our simulations find the refugee related casualty increases at the event level to be close to an average of the
province level refugee populations within affected provinces. This implies that, in the absence of additional data, applying a static severity increase in proportion to province migration statistics may be sufficient as a minimum estimate.

## 6  Conclusions

This study assessed the impact of Syrian refugee migration on earthquake casualty estimations in southeastern Turkey using a semi-empirical loss estimation technique on minimally modeled gridded population datasets created from refugee statistics and
Turkish ABPRS district level population data. It was shown that using the refugee adjusted population model in the earthquake fatality estimation process increased casualties in proportion to migrated population exposure. Earthquake scenarios on four of the five fault zones included in this study produced tens to hundreds of additional casualties after the inclusion of refugee data, with a maximum of 1579 extra casualties and a minimum of one extra casualty. While it naturally follows that places with population increases will sustain additional casualties in earthquakes, it had not yet been shown the degree to which current
refugee populations impact loss estimations.

Characterizing the expected casualty increases related to refugee migration is an important step in loss estimation—even if a basic province-level correction is a sufficient adjustment. Disaster scale underestimations have the potential to greatly complicate the work of local governments and aid agencies working to respond to earthquake disasters (Jaiswal et al., 2011b).



Accordingly, adjusting population models for refugee presence is an important consideration with casualty underestimations in high migration regions reaching hundreds of individuals. These adjustments will only increase in importance as more refugees flee into southeast Turkey alongside the evolving conflict in Syria. This study provided a methodology for making such adjustments in places where census data remains the de facto standard for environmental hazards studies.

This study incorporated refugees into earthquake loss estimations at the minimum possible level—considering them equal to that of local citizens. Further work improving the ability to characterize the seismic vulnerability of refugees is an important future step in understanding how their presence influences natural hazard evaluation. Changes in refugees' freedom of movement, reporting requirements, and settlement locations all affect the ability to uniquely incorporate them into earthquake risk analyses. In areas where the relative percentage of refugees amongst local populations continues to increase, understanding
where and how refugees are being accommodated should be a fundamental focus for earthquake related studies.

## 7   Data availability

The population models used in this project were constructed with freely available and frequently updated data from the address based population registration system (Turkish Statistical Institute, 2015) and the Turkish Ministry of Interior Directorate General of Migration Management (Republic of Turkey, 2015). The Global Earthquake Model's OpenQuake platform was used to
produce all earthquake simulations (GEM, 2016) in this study. The source models used as the basis for these simulations are available from the SHARE initiative (Giardini et al., 2013). Site amplification data used in scenario creation is available from the U.S. Geological Survey's global Vs30 grid (U.S. Geological Survey, 2013), described in Wald and Allen (2007). Building occupancy and collapse rate data from the WHE-PAGER phase I survey is published in Jaiswal and Wald (2009b). Please contact the corresponding author for the R loss estimation code or GIS processing workflows.

**Appendix A**

The process for determining fragility coefficients is described at length in Jaiswal et al. (2011a), with selected building types presented. A more complete list of coefficients was presented at the summer 2009 WHE-PAGER workshop (Jaiswal and Wald, 2009a).

*Acknowledgements.*  This research was performed under a National Science Foundation Graduate Research Fellowship, Grant: DGE-1450079.
We also thank the University of Arkansas for additional research support.



**Table 6.** Fragility coefficients

| PAGER-WHE Type | A | B | C | $R^2$ |
| --- | --- | --- | --- | --- |
| DS | 9.52 | -4.89 | 5.32 | 0.95 |
| A | 10.76 | -5.34 | 4.05 | 0.91 |
| UFB | 3.88 | -4.22 | 4.97 | 0.94 |
| UCB | 2.15 | -5.18 | 5.11 | 0.95 |
| C2 | 1.95 | -6.14 | 5.90 | 0.89 |
| C3 | 3.42 | -5.03 | 5.62 | 0.93 |
| C6* | 2.55 | -5.03 | 4.91 | - |
| C7* | 1.94 | -1.91 | 5.99 | - |
| PC2 | 0.85 | -2.35 | 5.90 | 0.95 |
| S1 | 0.45 | -8.71 | 4.40 | 0.80 |
| W6* | 1.14 | -2.66 | 5.49 | - |

$R^2$ denotes uncertainty compared to building performance records.
Asterisks indicate building types with fragility coefficients calculated
from a single expert estimate.

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
