# Peer review of "Assessing the impact of Syrian refugees on earthquake fatality estimations in southeast Turkey"

_Natural Hazards and Earth System Sciences, 2017_

## Referee Comment (RC1) · Anonymous Referee #1 · 18 Mar 2017

The paper introduces a model that is based on a number of assumptions that appear to be untested and are perhaps suspect. The first is that the migration data are accurate and representative of the situation on the ground. I doubt it, but what effort has been made to find out? The second is that there is no differentiation between the homes of the refugee populations and those of their local hosts. What if the refugees are concentrated into the most seismically precarious buildings? That is quite probable and it would inflate the death toll. Has this important point been investigated? The model produces a series of numbers which at the least represent spurious accuracy and may well simply mislead if the assumptions are inaccurate. Hence, something needs to be done to test them before the paper can be considered to offer a good

representation of the risks of seismic mortality.

The paper conflates 'casualty' and deaths. This is wrong: casualty refers to mortality and morbidity. It appears that the latter is not taken into account.

Pg 1, lines 24-25: "..high rates of urbanization have been attributed to the Turkish government's failure to enforce seismic building codes" - It seems odd to attribute urbanisation to failure to enforce codes. Vulnerability, yes, but urbanisation, I doubt.

Pg 2, line 3: "...to the disparaging earthquake mortality rates" - wrong choice of word

Pg 2, lines 24-25: "Because loss estimation systems can be used to direct rapid post-event humanitarian decisions..., their accuracy is crucial." - Not really. They are mainly used for actuarial purposes and some degree of approximate first estimation of effects, not for humanitarian decision-making.

This paper is quite sloppily written, with numerous errors of English.

---

## Author Comment (AC1) · 12 Apr 2017

Dear Reviewer,

Thank you for the feedback, we appreciate the comments on our manuscript.

The accuracy of migration data is a valid concern, and we have been actively seeking improvements. At this time however, we feel that the statistics we used are the best assumptions for refugee populations. They are reported as part of the Turkish Temporary Protection Regulations legal framework, and are consistent with refugee statistics reported by the U.S. Humanitarian Information Unit, the United Nations High Commissioner for Refugees, and other international agencies. Upon review, our discussion of

these statistics was vaguer than we intended and we will adjust these sections accordingly.

We included a discussion on the benefits, drawbacks, and methodology of our population model to facilitate an ongoing discussion in the scientific literature regarding uncertainties that do exist. As mentioned in the manuscript, most census based population models, even those disaggregated like GPWv4, do not include refugee populations from the Syrian conflict. Accordingly, we believe that our population model is more reflective of the situation on the ground than an unadjusted model. If an alternative approach, population model, or data source(s) are available, we would appreciate being pointed to specific publications or datasets. However, we have not come across such information in our review of related literature.

The critique on building classifications between refugees and local hosts is an important one, and something we would like to be able incorporate, as mentioned in the manuscript. However, we are unaware of any data that speaks in detail on the housing conditions of refugees in southern Turkey. As such, we selected an equivalent distribution to minimize our own speculation on the conditions and ensure our estimates are conservative. We do not intend to suggest that the situations of the two groups are equivalent, and will clarify our wording to articulate that death tolls could be higher in different housing conditions.

We also agree that there are inherent uncertainties in our numbers, as with any scenario-based loss estimations. The goal of the paper was to not to provide perfectly accurate death tolls, but to show that failing to include refugee populations in the calculations results in noticeable underestimations. This was the motivation behind concluding that a province-level severity adjustment may be a sufficient starting point. We will adjust our discussion to further emphasize the specific scope of our conclusions.

We appreciate the technical comments, and will revisit each of them in the manuscript.

We agree the urbanization statement was phrased awkwardly and will adjust it to reflect the original intent: high rates of urbanization have contributed to poor code enforcement. We have used the term casualty in the same context it is used in the literature outlining the semi-empirical approach within the USGS-PAGER system, the methodological basis for our estimations. However, we will review the manuscript and ensure its usage is consistent, along with other wording errors.

Best Regards, Bradley Wilson

---

## Referee Comment (RC2) · G. Papadopoulos (Referee) · 2 Sep 2017

Referee #2, Gerassimos A. Papadopoulos, Institute of Geodynamics, National Observatory of Athens, Greece

This is a well-written paper characterized by originality as regards the earthquake mortality assessment by taking into account population alteration due to massive refugee integration in the standard population of a country (here Turkey). Although the results

are susceptible to a number of assumptions and uncertainties, what is important with this paper is the methodological approach, not the results per se.The paper can be accepted for publication after minor improvements according to the next comments:

Figure 3, Legend. "The distribution of earthquake shaking as gathered from historical documents and modern seismic networks, compiled in (Sesetyan et al., 2013)". It should be "The distribution of earthquake epicenters as gathered from historical documents and modern seismic networks, compiled in Sesetyan et al. (2013)".

p. 6, l. 10 (and later): "...aeral distribution", means "...aerial distribution"? Please check. Table 1. Complete it by inserting information about earthquake magnitude, M, and fault length, L, for each fault segment. Explain how L is estimated from M. Table 2. Make reference to Table 6. p.12, l. 11-12: "These results follow out of Table 3 which shows populations generally transitioning from vulnerable masonry buildings to concrete structures during working hours". One may consider that during working hours the percentage of population being outdoors at a given earthquake time is higher than that during living hours. Possibly this could be included in the discussion. Appendix. I guess that Table 6 belongs to Appendix content. Make it more clear.

---

## Author Comment (AC2) · 2 Sep 2017

Dear Dr. Papadopoulos,

Thank you for the feedback and comments on our manuscript.

The problem in Figure 3, Legend is a LaTeX problem and easy to fix, thank you for spotting the error. Aeral distribution is a typo, it should be areal, we will fix accordingly.

Table 1: In the OpenQuake implementation we used for each scenario, a magnitude-area scaling relationship (Wells & Coppersmith, 1994) is employed within the scenario call to handle the magnitude scaling with the provided fault trace. We can explain in

[Figure]

greater detail how OpenQuake handles this process when discussion the scenarios. We did not include the magnitudes in the table because we had outlined that each fault was ruptured at the same three magnitudes, but can adjust the table if it is clearer with them included.

Table 2: Noted, will fix accordingly.

Table 6: We will work to clarify the relationship between Table 6, the text above, and other tables in the main text.

The point regarding indoor/outdoor population is an important one and we agree it should be included in the discussion, especially considering it has affected casualty totals in recent earthquakes (2015 Nepal comes to mind). Thank you again for the constructive suggestions for improvement.

Best Regards, Bradley Wilson

---

## Author Response (AR1)

Dear Editors & Reviewers:

We appreciate the constructive feedback on our manuscript. We have significantly edited our manuscript in response to reviewer comments. Particular focus has been placed on improving our discussion of relevant uncertainties and framing the results in the context of methodological limitations.

**Overview of Major Changes**

1. The title of the manuscript has been changed slightly to reflect the manuscript-level terminology change from 'casualty' to 'fatality'.

2. We have substantially edited the entire manuscript for clarity, consistency, and style. Special attention has been made to terminology concerns raised by reviewers, sources of uncertainty, and limitations.

3. We have introduced a major change in how building occupancy for refugees is calculated. The model now simulates a range of potential occupancy patterns for refugee populations to assess the magnitude of corresponding fatality variations. The related methodology, results, and discussion sections have been reworked accordingly.

4. The format of the discussion has been smoothed out. Fatalities are now discussed for non-refugee and refugee populations separately, then compared. Additionally, the comparison between refugee and non-refugee fatalities is now in the form of a new Figure instead of a table. We feel these changes better contextualizes the nature of the study.

5. Two subsections have been added to the discussion. One introduces the interpretation of fatality estimates and the other discusses total model uncertainty. We feel that that the addition of these two sections properly frames the limitations of our results.

**Response to Reviewers**

**Referee 1:**
*Migration data accuracy concerns:*
Concerns were raised over the migration data accuracy. We have included a more substantial explanation of where these statistics come from and what they cover. We have added a statement distinguishing between refugee populations and displaced persons, indicating that the data we used only covers registered refugee populations and may underestimate the total number of Syrians present in our study area. We have also improved the clarity regarding our disaggregation methodology and its relative strengths and drawbacks. (p. 6-7, section 4.1.1)

Additionally, it was not clear in our original draft that the statistics we were using are official registered refugee counts put out by the Turkish Department of the Interior---the same data used by the U.N. and U.S. Dept. of State in their publications. We have adjusted our manuscript to state this explicitly. (p. 6, l. 10-15)

*Differentiation between homes:*
We have completely redesigned how we account for the building occupancy of refugee populations in response to reviewer feedback. Refugee fatality estimations are now calculated 500 times, each with semi-randomly generated building occupancy percentages. The new methodology used to generate refugee occupancy tables is discussed at length in the new manuscript version, along with justification for the modified approach. (See p. 10)

We have updated the results to include the median fatality estimates along with median absolute deviations. We have also rewritten our discussion section for refugee related fatalities with more focus on the interpretation of the results. (See p. 13-14, Sec. 5.3)

*Casualties vs. Deaths*
Upon review of relevant literature, we determined that fatalities would be a more appropriate term than casualties. This has been changed throughout the entire manuscript.

*General Comments:*
Both grammatical errors pointed out have been fixed. (p. 2, l. 3-4, 8-9). The statement on loss estimations has been reworked to appropriately reflect the use of fatality estimation/rapid response systems. Three citations have been added to support our conclusions. (p. 2, l. 29-32.)

**Referee 2:**
*General Comments:*
Both errors pointed out by the reviewer (aeral instead of areal and improper citation format) have been fixed. (p. 5, Fig 3. & p.6 Section 4 – Data and Methods) Table 2 is now linked to the appendix when describing its construction (p. 10, l.1)

*Fault Parameter Table:*
We have added a more complete description of how the parameters in Table 1 are utilized in OpenQuake. As mentioned in our original response, OpenQuake uses magnitude and rake to create rupture areas within the calculations itself. Accordingly, changed the Table 1 title to 'earthquake rupture input parameters' to indicate that this information is being passed into OpenQuake. (p. 8, l. 7-17)

*Working hour population percentages:*
As recommended, we have referenced this in the discussion. We have also listed it as a limitation because our models do not account for outdoors populations. (p 12. l. 26)

[revised manuscript text omitted]

Dear Editors & Reviewers:

We appreciate the constructive feedback on our manuscript. We have significantly edited our manuscript in response to reviewer comments. Particular focus has been placed on improving our discussion of relevant uncertainties and framing the results in the context of methodological limitations.

**Overview of Major Changes**

1. The title of the manuscript has been changed slightly to reflect the manuscript-level terminology change from 'casualty' to 'fatality'.

2. We have substantially edited the entire manuscript for clarity, consistency, and style. Special attention has been made to terminology concerns raised by reviewers, sources of uncertainty, and limitations.

3. We have introduced a major change in how building occupancy for refugees is calculated. The model now simulates a range of potential occupancy patterns for refugee populations to assess the magnitude of corresponding fatality variations. The related methodology, results, and discussion sections have been reworked accordingly.

4. The format of the discussion has been smoothed out. Fatalities are now discussed for non-refugee and refugee populations separately, then compared. Additionally, the comparison between refugee and non-refugee fatalities is now in the form of a new Figure instead of a table. We feel these changes better contextualizes the nature of the study.

5. Two subsections have been added to the discussion. One introduces the interpretation of fatality estimates and the other discusses total model uncertainty. We feel that that the addition of these two sections properly frames the limitations of our results.

**Response to Reviewers**

**Referee 1:**
*Migration data accuracy concerns:*
Concerns were raised over the migration data accuracy. We have included a more substantial explanation of where these statistics come from and what they cover. We have added a statement distinguishing between refugee populations and displaced persons, indicating that the data we used only covers registered refugee populations and may underestimate the total number of Syrians present in our study area. We have also improved the clarity regarding our disaggregation methodology and its relative strengths and drawbacks. (p. 6-7, section 4.1.1)